# Productivity and Feed Quality Performance of Napier Grass (*Cenchrus purpureus*) Genotypes Growing under Different Soil Moisture Levels

**DOI:** 10.3390/plants11192549

**Published:** 2022-09-28

**Authors:** Ermias Habte, Abel Teshome, Meki S. Muktar, Yilikal Assefa, Alemayehu T. Negawo, Juarez Campolina Machado, Francisco José da Silva Ledo, Chris S. Jones

**Affiliations:** 1Feed and Forage Development, International Livestock Research Institute, Addis Ababa P.O. Box 5689, Ethiopia; 2Brazilian Agricultural Research Corporation (EMBRAPA), Brasilia 70770-901, Brazil; 3Feed and Forage Development, International Livestock Research Institute, Nairobi P.O. Box 30709-00100, Kenya

**Keywords:** Napier grass, Elephant grass, soil moisture stress, water use efficiency, feed quality, forage biomass

## Abstract

In the semi-arid and arid environments of Sub-Sharan Africa, forage availability throughout the year is insufficient and highly limited during the dry seasons due to limited precipitation. Thus, the identification of drought stress-tolerant forage cultivars is one of the main activities in forage development programs. In this study, Napier grass (*Cenchrus purpureus*), an important forage crop in Eastern and Central Africa that is broadly adapted to produce across tropical environments, was evaluated for its water use efficiency and production performance under field drought stress conditions. Eighty-four Napier grass genotypes were evaluated for their drought stress tolerance from 2018 to 2020 using agro-morphological and feed quality traits under two soil moisture stress regimes during the dry season, i.e., moderate (MWS) and severe (SWS) water stress conditions, and under rainfed conditions in the wet season (wet). Overall, the results indicated the existence of genotype variation for the traits studied. In general, the growth and productivity of the genotypes declined under SWS compared to MWS conditions. High biomass-yielding genotypes with enhanced WUE were consistently observed across harvests in each soil moisture stress regime. In addition, the top biomass-yielding genotypes produced the highest annual crude protein yield, indicating the possibility of developing high-feed-quality Napier grass genotypes for drought stress environments.

## 1. Introduction

Global warming and climate change have been described as potential threats to agricultural production and productivity due to increasing temperatures and declining and more erratic rainfall in the semi-arid and arid environments of Sub-Saharan Africa (SSA) [1]. The changes in precipitation and temperature levels trigger environmental stresses such as drought and heat stress that influence normal plant growth and hence minimize yield and quality. Forage crops are one of the feed sources for livestock production in SSA, but the availability of forage throughout the year is insufficient and much more limited in the dry seasons of the year due to inadequate precipitation [2]. Forage performance and production are highly associated with available soil moisture, so forage species adapted to semi-arid and arid environments need to be drought-tolerant and should offer a higher yield and feed quality potential to maximize the availability of feed in seasons with inadequate soil moisture [3,4]. Thus, the identification of drought- and heat stress-tolerant forage cultivars is one of the key strategies in forage development programs [5,6]. The evaluation of genotype performance under field conditions is one of the strategies for identifying tolerant and relatively productive genotypes in support of the development of varieties resilient to drought stress [7]. The morphological and physiological traits of plants are very important for selection to improve drought tolerance due to their relation to the adaptation of future climate scenarios.

Napier grass, or Elephant grass (*Cenchrus purpureus* (Schumach.) Morrone syn. *Pennisetum purpureum* Schumach.), is an important forage grass in tropical and sub-tropical environments, growing mainly from sea level up to 2000 m.a.s.l., with the best growth at temperatures ranging from 25 to 40 °C [8,9]. Napier grass is also considered a short-term drought-tolerant forage, an important characteristic in areas that frequently face drought stress conditions [10]. It is a popular forage crop in tropical environments, mainly due to its high biomass production per unit area when compared with other tropical forages such as Guinea grass and Rhodes grass [11]. The International Livestock Research Institute (ILRI) forage gene bank conserves a range of Napier grass genotypes that are comprised of collections from ILRI and the Brazilian Agricultural Research Corporation (EMBRAPA) [11,12]. Diversity studies conducted on these collections have revealed the existence of genetic variation with the potential to support the development of varieties with desirable traits suitable for forage production in SSA. Phenotypic and genotypic diversity studies conducted on these collections revealed the existence of genetic variation with the potential to support sustainable forage production in the region and subsequently contribute towards improved animal performance [12]. Likewise, the field characterization of those collections indicated the existence of genotypic variations for yield and feed quality traits under both wet and dry season growing conditions, further indicating the possibility of selecting genotypes that fit the trending climatic conditions [13]. Generally, the high biomass-yielding capability and adaptability to wider environments of Napier grass make it an alternative forage species for livestock feed in moisture stress environments. Many cultivars of Napier grass, and hybrids with pearl millet (*Cenchrus purpureus* (Schumach.) Morrone × *Cenchrus americanus* (L.) Morrone), have been developed in different parts of the world for their high yield potential, broad adaptability, resistance to diseases and feed quality traits [14]. Despite the information on the adaptability of Napier grass to short-term drought stress, there have been no genotypes identified that can tolerate and produce biomass to support the availability of feed for livestock in the longer dry season conditions of tropical environments. Thus, the present study was conducted to evaluate Napier grass genotypes for enhanced water use efficiency (WUE) under field drought stress conditions using different soil moisture regimes in the dry season, from 2018 to 2020. Overall, the results of the analysis of agro-morphological and feed quality traits indicated the existence of significant phenotypic diversity among the experimental genotypes. Consistently high biomass-yielding genotypes with enhanced WUE were observed across all harvests in each soil moisture regime in the dry season. Furthermore, the additive main effects and multiplicative interaction (AMMI) analysis identified productive and stable genotypes which were consistent with the identified productive genotypes.

## 2. Results

### 2.1. Effects of Growing Season, Harvest Period and Genotype on Napier Grass Performance

To analyze the performance of Napier grass genotypes, 12 traits representing morphological, physiological and agronomic parameters (Table 1) were investigated from the plants grown under moderate water stress (MWS) and severe water stress (SWS) conditions in the dry season as well as in the main rainy season (wet). The principal component analysis, conducted using all trait values from all growing conditions and harvests, showed a clear separation of genotype performance between the dry and rainy seasons, with relatively little variation in agronomic or feed quality traits between the MWS and SWS conditions (Figure 1). Furthermore, analysis of variance (ANOVA) indicated highly significant (<0.001) variation among genotypes and harvests in each of the dry season and wet season conditions for all the traits considered, while the difference between treatments (MWS and SWS) was only highly significant (<0.001) in the dry season harvests (Table 1). Regardless of the production season, the interaction effects among different genotypes and harvests were highly significant (<0.001) for all traits; however, the genotype-by-treatment effect was only significant in the dry season. This finding indicates that the plants grown under the SWS condition recovered more quickly with the onset of the rainy season. This recovery potential is an important response to determine the overall growth and development performance of the genotypes. 

In addition, a separate ANOVA conducted for the dry season revealed significant (<0.05) and highly significant (<0.001) differences among genotypes, treatments and harvests, as well as in the interaction effects, for all traits except for leaf width (LW) and chlorophyll fluorescence (Fv/Fm) (Table A2), but in the rainy season, only the genotype and harvest effects were statistically highly significant (<0.001) (Table A2). Overall, the results showed that the performance of Napier grass was primarily affected by the genotype and harvest round in each of the wet, MWS and SWS growing conditions. The observed genotypic variations are important to exploit the potential of the genotypes to maximize forage production under different water stress environments.

### 2.2. Partitioning Quantitative Genetic Variation

The phenotypic coefficient of variation (PCV), genotypic coefficient of variation (GCV) and broad sense heritability (H^2^) were calculated to assess the contribution of these factors to the respective traits (Table 2). Generally, the PCV values are greater than the GCV values for all growing conditions, a finding that indicates that both genotypes and environmental factors contributed to the observed variation in the traits under investigation. The PCV and GCV values under the wet condition were higher than those under the dry season treatments (MWS/SWS) for the corresponding traits. In the wet condition, the highest PCV and GCV values were recorded for LSR (115% and 69%, respectively), followed by TFW (89% and 55%, respectively) and TDW (82% and 59%, respectively). In the wet condition, the PCV values for LSR, PH and IL were approximately two-fold higher than the GCV values, indicating that the environment plays a significant role in the variation observed among genotypes for these traits.

The PCV and GCV values were similar to each other for the corresponding traits in the MWS and SWS conditions, which suggests that the effects of genotype and environmental factors on trait expression in both dry season treatments were similar. The traits with the highest PCV values, when growing under the MWS and SWS conditions, were WUE (80% and 89%, respectively), TFW (69% and 71%, respectively) and TDW (68% and 70%, respectively), and the corresponding GCV values for these traits when growing under these conditions were (39% and 41%, respectively), (41% and 42%, respectively) and (56% and 43%, respectively). 

The broad sense heritability estimates were higher for LW, TN, TFW and TDW when growing under the wet condition than those under both the MWS and SWS conditions, while the heritability for PH and LL was lower under the wet condition than that under both the MWS and SWS conditions. The heritability estimates of the corresponding traits were similar for the MWS and SWS conditions (Table 2).

### 2.3. The Effect of Soil Moisture Stress Levels on Napier Grass Performance

The current study revealed that soil water stress conditions had a different effect on the physiological parameters, such as water use efficiency (WUE), as the genotype mean values for SWS were higher than the values for the corresponding genotypes grown under MWS. In addition, this observation showed the capability of different genotypes to show enhanced water use efficiency when exposed to different soil moisture stress conditions. Under MWS, the highest WUE was observed for genotypes 16819 (3.94), CNPGL 92-66-3 (3.73) and BAGCE 30 (3.72), and under SWS, the highest WUE was observed for genotypes 16819 (4.12), 16802 (4.1) and CNPGL 92-66-3 (3.98).

The mean trait values for wet, MWS and SWS conditions for 12 agro-morphological and physiological traits are presented in Appendix A. The genotype values of all traits decreased under the dry season treatments compared to the rainy season conditions, except for tiller number (TN). The physiological parameters that indicate photosynthetic efficiency, Fv/Fm and PI, were lower in the dry season (MWS and SWS) for corresponding genotypes, indicating that the imposed field water stress treatments impaired the normal physiological function of the plants. The lower Fv/Fm and PI mean values for genotypes growing under SWS compared to the corresponding genotypes growing under MWS support the assumption that the water stress severity under SWS was higher than that under MWS. This difference in severity level was shown by the lower mean values for genotypes grown under SWS, except for LW and LL, than the corresponding genotypes growing under MWS. However, the degree of decline was different for the various genotypes and at different harvests, as revealed by the genotype-by-treatment interaction effects (Appendix A).

A stress tolerance index (STI), one of the most commonly used drought stress tolerance indices [15], was used to estimate the tolerance levels of the genotypes when growing under both MWS and SWS conditions. Generally, the STI value was lower for genotypes growing under SWS conditions compared to the corresponding genotype growing under MWS conditions (Appendix A), a finding that reflects that the more severe the treatment conditions, the greater the influence on genotype performance. Under MWS, the highest STI was observed for genotypes 16819 (0.53), 16791 (0.46) and BAGCE 30 (0.45), and under SWS, the highest STI was observed for genotypes 16819 (0.49), 16791 (0.47) and CNPGL 93-37-5 (0.41). In addition, this observation showed the capability of different genotypes to show enhanced water use efficiency when exposed to different soil moisture stress conditions. Under MWS, the highest WUE was observed for genotypes 16819 (3.94), CNPGL 92-66-3 (3.73) and BAGCE 30 (3.72), and under SWS, the highest WUE was observed for genotypes 16819 (4.12), 16802 (4.1) and CNPGL 92-66-3 (3.98).

The WUE and STI profiles for Napier grass genotypes growing under both MWS and SWS conditions are presented in Figure 2. Regardless of the treatment under dry season conditions, genotypes showed a similar performance trend for both WUE and STI, indicating the potential importance of these traits for the screening and identification of genotypes for drought stress tolerance.

### 2.4. Genotype Diversity and Trait Selection under Water Stress Conditions

To evaluate the diversity of genotypes and select traits for both MWS and SWS growing conditions, the morphological, physiological and agronomic traits were subjected to principal component analysis (PCA). In both stress conditions, the first two principal components (PCs) explained approximately 80% of the genotype variation (Figure 3). Under MWS, PC1 explained about 62% of the total variation, and this was principally associated with the traits STI, TDW, TFW, WUE and LL. PC2 under MWS explained about 18% of the total variation and was mostly associated with the traits PI, TN and Fv/Fm (Figure 3A and Table A3). Similarly, under SWS, PC1 explained about 61% of the total variation and was associated with STI, TDW, TFW, WUE and LL, and PC2 explained about 19% of the total variation, mostly associated with PI, Fv/Fm and TN (Figure 3B). Overall, the first principal component is mainly related to forage biomass and architecture traits, while the second principal component is related to physiological parameters that are indicative of photosynthetic efficiency. 

### 2.5. Genotype and Trait Clusters under Water Stress Treatments

Biplot and hierarchical cluster analyses were conducted separately for both MWS and SWS conditions using TDW, WUE and STI to partition genotypes based on their drought stress response (Figure 4 and Figure 5). Under both MWS and SWS conditions, the first two principal components (PC1 and 2) explained more than 90% of the genotype variation (Figure 4A and Figure 5A). The PCA categorized genotypes into highly drought stress-tolerant (high-yielding and water use-efficient), moderately tolerant and susceptible genotypes. Genotypes that were on the positive side of PC1 were high in WUE, TDW and STI, while genotypes on the negative side were low for the respective traits. Similarly, the hierarchical clustering categorized genotypes into highly tolerant, moderately tolerant and susceptible clusters in both soil moisture stress conditions. Furthermore, the drought stress response grouping of genotypes in both moisture stress conditions revealed that the highly tolerant and susceptible genotype groups were similar and showed the consistent performance of these genotypes regardless of the drought stress conditions (Figure 4B and Figure 5B). The performance of the moderately tolerant genotypes varied depending on the soil moisture stress levels.

In both stress levels, genotypes 16819, 16791, BAGCE 30, CNPGL 92-66-3 and BAGCE 93 were considered highly tolerant to drought stress, with consistently high TDW and WUE performances in both conditions, while genotypes 16834, 16805, 16621, 16790 and 16797 were the most susceptible. 

### 2.6. Biomass Productivity of Napier Grass Genotypes under Water Stress and Optimum Soil Moisture Conditions

To assess the biomass productivity potential and maximize the availability of feed across seasons, genotypes grown under dry (MWS and SWS) and wet season conditions were compared by assessing TDW production. The top biomass-yielding genotypes identified in the wet season were 16791, 16819, BAGCE 30, CNPGL 93-37-5, similarly 16819, 16803, 16839 and BAGCE 30 under MWS and 16819, CNPGL 93-37-5, CNPGL 92-66-3 and 16839 under SWS (Appendix A). Furthermore, a general linear model regression between rainy and dry (MWS and SWS) season conditions for TDW revealed a positive correlation between rainy and MWS (R^2^ = 0.53) conditions and between rainy and SWS (R^2^ = 0.66) conditions (Figure A1A,B). This result suggests that the high potential yield under optimal conditions could also result in improved yield under water stress conditions.

### 2.7. Biomass Yield Stability across Harvests

To improve the efficiency of selecting the best genotypes for wet, MWS and SWS conditions, Additive Multiplicative Interaction (AMMI) analysis was employed for TDW, where the sum of squares for the yield was partitioned into genotype (G), environment (E) and GE interaction (GEI), where the GEI was further partitioned by the interaction of principal components (IPCs) (Table A4). The results of the AMMI analysis indicated a significant GEI effect, and further partitioning of the total sum of squares revealed that the genotype effect accounted for the largest proportion of the sum of squares for TDW, followed by E and then GEI, under both MWS and SWS conditions, while in the wet condition, the environment factor was the most significant, followed by genotype and GEI. The first two IPCs contributed the highest GEI partitioning for all growing conditions.

The AMMI stability value (ASV) ranked the genotypes based on their score, with the lowest scores representing the most stable genotypes (Table A5). Under the wet condition, the most stable genotype for TDW yield was CNPGL 96-27-3, followed by 16789 and 16821, which had ASV rankings of 1, 2 and 3, respectively, while the least stable genotype was 16819, followed by 16791 and 16621, ranking 84, 83 and 82, respectively. Under MWS, the most stable genotypes for TDW yield were 15743, followed by 16806 and 16803, while the least stable genotype was 16839, followed by BAGCE 30 and CNPGL 93-37-5. Under SWS, the most stable genotype for TDW yield was 16792, followed by 15357 and 14389, while the least stable genotype was 16839, followed by BAGCE 93 and BAGCE 30. Thus, high-yielding genotypes were identified as less stable and vice versa based on ASV analysis. To include productivity as a selection criterion for drought stress tolerance, yield stability index (YSI) was used, as it encompasses the rank from both ASV and overall yield. The genotypes that had low YSI scores under the wet condition include CNPGL 92-66-3, 16839 and 14983, while for the MWS condition, genotypes 16803, 16806 and 14984 had the lowest scores; similarly, under SWS, genotypes 16792, CNPGL 92-66-3 and 14389 had the lowest YSI score. 

### 2.8. Feed Quality Trait Variation among Genotypes, Soil Moisture Levels and Harvest

The combined ANOVA under wet, MWS and SWS conditions revealed highly significant genotypic and harvest round differences for feed quality traits (NDF, ADF, ADL, OM, CP, IVOMD and Me) (Table 3). The observed differences between the genotypes, harvest round and genotypes × harvest indicated that the studied feed quality traits were greatly influenced by both genotype and harvest round (Table A6).

The PCV, GCV and broad-sense heritability (H^2^) for feed quality traits are presented in Table 4. The mean values for the fiber components NDF, ADF and ADL were reduced in moisture stress conditions (MWS and SWS) compared to the wet condition; conversely, the mean value for CP, IVOMD and Me increased under moisture stress conditions (Appendix A). Similarly, a radar plot shows that almost all genotypes studied had reduced levels of fiber components under the stress conditions (Figure A1A–C), while the levels of CP, IVOMD and Me increased (Figure A1D–F). The PCV values were higher than the respective GCV values in all soil moisture conditions, indicating that the environment has a considerable effect on the feed quality traits (Table 4). Generally, PCV values were highest under the SWS condition, followed by the MWS and wet conditions, except for NDF and ADL, suggesting that the environmental impact on the expression of feed quality traits increased as soil moisture levels decreased. The maximum estimated value for PCV and GCV was for CP, followed by ADL and ADF, whereas the minimum estimated values were for OM and NDF. Heritability is a measure of the degree of influence of genotype and environment on the expression of the parameters. In our study, the estimated heritability values were medium to low; the heritability estimates for feed quality traits were lower in the SWS condition, except for NDF and ADL. Overall, the highest heritability was observed for OM and NDF for the studied genotypes.

### 2.9. Association of Feed Quality Traits and Total Dry Weight under Moisture Stress Conditions

The principal component analysis (PCA) showed that the first two principal components (PC) for MWS and SWS conditions accounted for 68.3% (44.4 and 23.9%) and 71.5% (49.3 and 22.2%) of the variation, respectively (Figure A2). PC1 was positively correlated with ADF, ADL and NDF and negatively correlated with CP, IVOMD and Me. PC2 was mainly positively correlated with OM and NDF. The PCA further revealed the association between different feed quality traits and genotypes, as demonstrated by the PC biplots (Figure A2A,B) for both water stress conditions. A smaller angle between different feed quality traits in the same direction indicated a high association between corresponding traits for classifying genotypes. Under both soil moisture stress conditions, the variations in IVOMD, CP, Me and ADF were the biggest contributors to PC1, while the variances for OM and NDF were the biggest contributors to PC2. Along PC1, genotypes located on the right side of the origin had high values for IVOMD, CP and Me, while the genotypes located on the left side had low values for these traits. Similarly, along PC2, genotypes at the top of the origin had high values for OM and NDF, while genotypes at the bottom revealed low values for these traits. Genotypes superior for a particular parameter were located close together and along the same direction of the vector line. 

### 2.10. Annual Total Dry Weight and Crude Protein Yield Performance of Genotypes

To assess the forage value of genotypes, cumulative annual biomass and crude protein yield were evaluated. The range of annual TDW for the MWS condition was between 2.65–68.05 t/ha/year, with a mean value of 34.14 t/ha/year (Table 5). The TDW range for the SWS condition was between 2.47–67.35 t/ha/year, with a mean value of 32.95 t/ha/year. Genotypes 16791, 16819, BAGCE 30, CNPGL 93-37-5 and 16802 produced the most annual TDW in both conditions. The genotypes that produced the highest annual TDW also produced the highest annual crude protein yield (CPY).

### 2.11. Variation between Leaf and Stem Tissue Samples for Feed Quality Traits in the Wet Season

The combined ANOVA of feed quality traits for leaf and stem tissue samples under the wet condition is presented in Table A7. All the studied feed quality traits showed highly significant genotypic and harvest round variation, while prior moisture stress treatments (MWS and SWS) in the dry season did not appear to affect them.

Leaf and stem tissue samples differed in their feed quality traits; generally, leaf samples had higher CP levels, while stem samples had higher fiber components (NDF and ADF) (Figure 6 and Table A8). Interestingly, the study revealed genotypes that had stem samples with a high IVOMD and leaf samples with high ADL (Figure 6A,B). 

Among the genotypes, 16811, CNPGL-93-01-1,16790 and BAGCE 86 had high amounts of CP in the leaves, while 16819, 16836 and 16812 had high CP in the stems. Genotypes 16794, BAGCE 100 and 1026 had high fiber components (NDF, ADF and ADL) in the stems, while 16786, BAGCE 30, BAGCE 34 and 16797 had high leaf fiber components (Figure A4A,B; Appendix A).

Neutral detergent fiber (NDF); acid detergent fiber (ADF); acid detergent lignin (ADL); organic matter (OM); crude protein (CP), in vitro organic matter digestibility (IVOMD); metabolizable energy (Me). Values indicate the significance probability level; not significant (NS).

## 3. Discussion

Drought stress is one of the main challenges that influence forage production in the arid and semi-arid tropics, particularly in the dry season. Developing forage cultivars that are high biomass-yielding and drought-tolerant is a priority in the tropical regions where droughts are increasing. Generally, biomass yield is determined by overall plant growth and developmental processes; thus, plant growth and development attributes are used to screen for drought adaptability. Traditionally, forage breeders utilize morphological and physiological indicators to screen genotypes for drought tolerance, often coupled with feed quality analyses. Phenotypic assessments based on adaptive morphophysiological—including yield-related and feed quality—traits were the key indicators to identify drought-tolerant, productive and higher-quality forage genotypes [16,17]. Napier grass is one of the best adapted tropical forage species that can withstand drought stress [18,19]. In this study, genotypes from the ILRI and EMBRAPA collections were used to identify genotypes better adapted to produce under soil moisture stress conditions. The present study examined the performances of 84 Napier grass genotypes for morphophysiological, agronomic and feed quality traits under moderate and severe water stress conditions in the dry season, as well as under rainfed conditions in the wet season. In our study, the field drought stresses (MWS and SWS) in the dry season altered the performance of morpho-agronomic and feed quality traits in the crop compared to plants grown during the rainy season. Both MWS and SWS conditions negatively influenced the plants’ morphological development by decreasing PH, LL and LW in all of the studied Napier grass genotypes, since optimum soil moisture is important for nutrient uptake and plant growth and development [20]. However, the number of tillers did not decrease under drought stress treatments. This may be attributed to the adaptive nature of tiller density, which indicates increased resource use efficiency under stress conditions, enhancing the chances of survival and fast recovery [21,22]. Such performance changes have also been noted in previous studies that reported on the morphological and physiological adjustments of Napier grass plants when exposed to drought stress conditions [19]. The reduction in morphological development could be associated with a decline in cell growth and expansion that limits the overall plant architecture when growing under drought stress. Similarly, drought stress affected the photosynthetic efficiency (Fv/Fm and PI) and forage biomass yield (TFW and TDW) compared to growth in the rainy season. Thus, the reduction in photosynthesis may decrease assimilate accumulation, which ultimately determines biomass yield. Taken together, the observed changes in growth performance are related to differences in soil moisture levels between the rainy and dry seasons [13]. This underscores the fact that soil moisture is one of the driving factors in determining the overall growth and development of Napier grass. 

Generally, highly significant genotypic differences were observed for all the studied morpho-agronomic and feed quality traits in both rainy and dry season conditions, indicating that the tested genotypes had inherent genetic differences for the studied traits. Genotype × harvest interaction was also significant for TDW, indicating that TDW was quantitatively inherited and differentially expressed in response to different harvest rounds. 

### 3.1. Trait Expression under Rainfed, Moderate and Severe Water Stress Conditions

In the dry season, there was significant variation between drought stress treatments (MWS and SWS) for most agro-morphological and feed quality traits (Table 1). Genotype-by-drought-stress-treatment interactions were also significant for most traits, indicating that genotypes performed differently under MWS and SWS conditions. Thus, independent selection for MWS or SWS conditions can be used as a strategy to identify the best performing genotypes, to exploit their potential and to subsequently maximize forage production in the dry season with minimal irrigation support. 

In the rainy season, genotypes showed significant genotypic variation for morphological, agronomic and feed quality traits (Table 1 and Table 4). These phenotypic variations between genotypes are indicative of the opportunity to select genotypes for increased performance under optimum soil moisture conditions. However, under the wet season condition, there was no statistically significant difference for all morpho-agronomic traits between the blocks that were treated with either MWS or SWS in the dry season. For example, in the wet season, we detected no significant difference in total dry weight production between genotypes exposed to MWS and SWS conditions in the dry season. However, the reduction in biomass yield was higher under SWS than it was under MWS, showing that severe water stress had reduced the biomass yield. Thus, the similar annual above-ground biomass production could be attributed to an enhanced growth rate stimulated by the increased assimilation rate of genotypes grown under SWS. The re-allocation of stored carbohydrates in the root may have contributed to the stronger compensatory growth of plants exposed to severe water stress [23]. This indicates that genotypes which were grown under the SWS condition in the dry season had a greater recuperative ability during the rainy season when the soil moisture became optimum. Hence, such recovery potential is an important response to determine the overall productivity performance of Napier grass genotypes for tropical environments with a long dry season and a short rainy season. 

Furthermore, the feed quality traits in the rainy season were not affected by the preceding drought stress treatments (MWS and SWS) in the dry season (Table 3). Since feed quality traits are strongly associated with plant development, the similarities displayed in the morphological and agronomic performance of the plants between treatments might have also resulted in similarities in feed quality status. The results also indicated that significant variation existed between harvests for the quality traits considered, which could predominantly be influenced by the soil moisture conditions due to the differences in the received precipitation during growth periods in the wet season (Table A6).

The present study revealed different levels of variation and heritability among genotypes for all traits measured. Higher levels of variation and heritability were observed during the rainy season compared to the dry season for TN, TFW and TDW traits. These traits recorded the highest values for PCV and GCV in all conditions, indicating the presence of high genetic variation. In the drought stress treatments, TDW, WUE and TN had the highest PCV and GCV values, indicating the presence of high genetic variation, while Fv/Fm and LW had low PCV and GCV values, indicating low genetic variation under both MWS and SWS conditions. Thus, the adaptive traits with high amounts of variation under water stress treatments are worth considering for the effective screening of genotypes for performance under drought stress. The observation of higher PCV than the respective GCV values indicates the significant contribution of environmental effects on trait expression. On the other hand, the high heritability estimates for LL, TN, LW and TDW indicate that the trait expression was predominantly governed by additive gene actions, so direct selection might be effective to improve these traits. Our findings concur with previous studies that reported considerable variation in plant growth traits in perennial forage grasses [24]. Subsequently highly heritable traits, such as LW and LL, can be efficiently used to support improved yield in Napier grass [25]. Under drought stress treatments, strong and positive associations were observed between biomass yield and plant growth and development traits (PH, LL, LW and TN). These morphological traits can play an important role in adaptation under soil moisture stress conditions; thus, the indirect selection for biomass productivity based on these morphological traits could be effective [26]. Plant species usually implement multiple drought resistance strategies, varying morphological and physiological traits in response to drought stress [27]. The present study demonstrated that, under both levels of soil moisture stress, the most productive genotypes generally have superior plant height, leaf length and leaf width and increased tillering compared to less productive genotypes. Therefore, as an adaptation strategy, we suggest that the drought-tolerant genotypes have the ability to enhance post-stress regrowth [28].

In contrast to the agro-morphological traits, the level of variation and heritability was generally low for feed quality traits. In all conditions, relatively high amounts of variation and heritability were observed for the traits CP, ADL and ADF. Generally, PCV values were more than two-fold the respective GCV values, indicating that environmental effects strongly influenced trait expression. 

### 3.2. Drought Stress-Responsive Agro-Morphological Traits

A PCA analysis was conducted to explain and describe important indicators of stress resistance in the plants that can be utilized for the selection of drought-tolerant genotypes. The PCA indicated that, under drought stress treatments, STI, WUE, PH, LL and LW significantly influenced biomass yield (Table A3). In both drought stress treatments, the PCA biplot revealed that most of the variation was contributed by PC1, where WUE and STI accounted for the highest positive contribution, whereas Fv/Fm and PI contributed negatively to the variation. Thus, the major positive contributing traits can act as a promising indicator for screening Napier grass genotypes growing under moisture stress conditions. Previous studies revealed that WUE is an important characteristic of a plant’s response to drought stress, where a high WUE indicates adaptability to a drought environment [27]. The biplot created between PC1 and PC2 showed the grouping of genotypes along the vector line. Genotypes 16839 and 16801 were located at a considerable distance along the line from the origin and hence can be considered drought-tolerant genotypes. The PCA biplot also showed that genotypes 16839, 16791 and 16819 were located close to the vector line that is associated with drought-tolerant indicators (WUE and STI); thus, these traits could be employed for identifying groups of Napier grass genotypes with drought stress tolerance and susceptibility. These findings indicate that the maintenance of high WUE and STI makes the greatest contribution to biomass yield compared with other morphological traits. 

### 3.3. Drought Stress Effects on Genotype Performance

The evaluation of genotypes growing at different drought stress severity levels is useful to support the selection of high-yielding and stable genotypes for production in drought-prone environments. The current study indicated that the SWS condition resulted in a greater reduction in the growth and development of most genotypes compared to the MWS treatment. Plants utilize various adaptation mechanisms to cope with different drought stresses and ensure growth and development. The present study revealed that Napier grass genotypes are able to survive under reduced soil moisture conditions, as no genotype perished under the SWS condition, although growth and development were negatively affected. The observed genotypic variation for WUE and STI under both MWS and SWS conditions highlights the potential to screen Napier grass genotypes for enhanced drought stress tolerance. The genotypic variation in WUE could be linked to a reduction in stomatal density and leaf width and length so that the plant maintains its internal water balance [29] or increased water status due to the increased root depth and biomass under drought stress [3,4]. These suggestions may be the subject of future research. 

A PCA identified that enhanced WUE, STI and TDW were the key indicators for the analysis of the response of genotypes to drought stress. Accordingly, these traits were able to distinguish drought-tolerant and -susceptible genotypes maintained under both MWS and SWS conditions. In both stress conditions, genotypes were distinctly categorized into highly tolerant and susceptible groups, while some genotypes that were grouped in the highly tolerant group under MWS were grouped in the moderately tolerant group under SWS. Among the genotypes, 16819, 16791, BAGCE 30 and 16839 were considered to be highly drought-tolerant. On the other hand, genotypes such as 16834, 16790, 16797 and 18662 were found to be susceptible. Our findings confirmed a previous report that indicated that genotype 16790 was a low-performing genotype, while genotype 16791 was highly productive [30]. 

### 3.4. Biomass Yield of Genotypes

The evaluation of the total dry weight production of genotypes across the wet, MWS and SWS conditions revealed that the highly productive genotypes identified under the rainy season were also high yielders under both MWS and SWS conditions. A similar observation was also reported by [13], where the top yielding genotypes on an annual basis also performed best under dry season production conditions. Furthermore, the general linear model regression for TDW between rainy and MWS conditions and between rainy and SWS conditions also substantiated the observation of consistent, productive genotypes across the seasons and drought stress treatments. The highly productive genotypes during the wet season include 16791, 16819, BAGCE 30, CNPGL 93-37-5 and 16802. Our findings align with a previous study that reported that genotypes 16791 and 16819 were high biomass yielders, and these accessions have already been released as commercial cultivars in Ethiopia [21]. Under MWS, genotypes 16819, 16803, 16839, BAGCE 30 and 16811 produced the highest forage yield. Similarly, under SWS, genotypes 16819, CNPGL-93-37-5, 16839, BAGCE 100 and BAGCE 30 were the top forage producers. Therefore, we suggest that the selection of Napier grass genotypes under optimal and water stress conditions is a suitable approach to obtain genotypes that can reliably produce high biomass yields in the dry season and maximum yields in optimum soil moisture conditions. Regarding the annual biomass production, the highest dry biomass yield was obtained from highly productive genotypes identified under wet season and drought stress conditions (Table 5). These top biomass-yielding genotypes were tall [31] with low LSR values, indicating that they had a high stem biomass, although this could, at least in part, be associated with their accelerated development due to their high production. A separate study that agreed with our findings showed that WUE was higher in vigorous genotypes than it was in low-yielding genotypes [30]. The top biomass-yielding genotypes could: (a) avoid drought stress by regulating water loss or improving water uptake, or (b) store water in the stem [32], thereby enhancing WUE [4]. Other reports also suggest that productive genotypes have the capacity to compensate for the reduced soil moisture by the enhanced recovery potential in the rainy season, whereas less productive genotypes are slow-growing in addition to being sensitive to the stress that further limits plant development [33]. 

The present study revealed significant genotype-by-harvest-round interaction in each of the three growing conditions, as indicated by the crossover performances for genotypes on different harvest rounds using AMMI analysis (Table A4). This might lead to variations in the mean ranks of the genotypes at different harvests. Thus, AMMI stability value analysis was performed to identify genotypes whose forage biomass yield fluctuated less over the seasons. Here, genotypes with low ASV values are scored as highly stable, whilst those with high ASV values are scored as less stable. However, ASV stability alone for biomass yield might not indicate productivity because consistently low-yielding genotypes, such as 16790 and 16621, have been shown to be highly stable. Therefore, the yield performance index (YSI), the sum rank of the mean yield across the time of harvests with the rank of the ASV of genotypes, was employed to determine both stable and productive genotypes. Genotypes with a low YSI are considered for selection, as they combine a high yield and stable performance. The results of the present study showed that high-yielding genotypes such as 16791, 16819, BAGCE 30 and BAGCE 100 had high ASV values and were categorized as moderately to lowly stable genotypes using the yield stability index. A similar observation has been reported by a previous study [21], where high-yielding genotypes, such as 16791, had low stability [34]. 

### 3.5. Feed Nutrient Quality Performance

The feed quality of forage is an essential component to consider during selection. Significant genotypic variation for feed quality traits was observed in each of the wet, MWS and SWS growing conditions, suggesting the potential to select for these traits. This study revealed the existence of genotypic variation for feed quality traits, similar to previous reports [35,36]. In addition, the growing conditions influenced the feed quality; for example, the mean values of NDF, ADF and ADL were higher during the rainy season, while CP, IVOMD and ME were high in the dry season [37]. The decrease in CP and IVOMD during the rainy season may be associated with a dilution effect due to increased phenological development. The observation of reduced NDF in the dry season was also reported by a previous study [30]. However, in contrast to these studies, another study [21] reported increased NDF in the dry season, although these conflicting observations could be associated with differences in the maturity level at harvest between the two trials [33]. Our findings have revealed that genotypes that produced the highest CP content were consistent across the three growing conditions, indicating the inherent characteristics of the genotypes. As expected, the fiber components (NDF, ADF and ADL) were negatively correlated with CP, IVOMD and Me [37].

The combined biplot analysis of Napier grass genotypes across the two drought stress treatments identified clusters of genotypes with high fiber components (NDF, ADF and ADL) such as 16839, BAGCE 100 and 16819 [34] or high CP and IVOMD, such as 16811, BAGCE 81 and BAGCE 17 (Figure A2). These results indicate that there is considerable opportunity for the improvement of Napier grass in terms of different forage quality traits. Genotypes that produced a high biomass yield under MWS and SWS also had a higher fiber content, presumably to support increased growth, while their CP content was low. A similar result was also reported in a previous study [36], where high dry matter-producing Napier grass cultivars exhibited low nutritional quality. However, the present study showed that the CP content of the productive genotypes was more than the 7% that is the minimum required level for rumen microbial activity [38]. It is intriguing to note that genotypes that combined both a good feed quality and a high biomass yield, such as 16811 and 14983, were identified. 

Leaf-to-stem ratio (LSR) has been shown to be a key component in determining feed quality. In the rainy season, the highest LSR was observed in genotypes 18448, BAGCE 17, 16902 and 16787—genotypes that also had high CP and IVOMD values. The high biomass-yielding genotypes 16791, 16819 and BAGCE 100 had low LSR values, while their fiber components were high. Nevertheless, considering the annual CP yield, these high biomass-yielding genotypes produced the highest CP yield, suggesting that annual biomass production coupled with annual feed nutrition values such as CP yield are important traits as far as forage value is concerned. 

Partitioning the forage value into leaf and stem fractions in terms of nutritional composition is a key breeding strategy to develop higher-quality forages. Generally, the leaf is more digestible, higher in CP and lower in fiber components than the stems [39]. Such differences in feed nutrition between the leaf and the stem might be attributed to the photosynthesis/metabolic role of leaves and the structural role of stems [40]. Consequently, improving feed quality relies, at least partially, on increasing the proportion of the leaf when harvested. Consistent with previous reports, our findings show that the leaves generally have higher CP and IVOMD values and lower fiber components than the stems [36,41]. However, it is interesting to note that the stem tissues of some Napier grass genotypes have high CP and IVOMD values compared to the leaves. This may provide a biochemical basis for the selection of Napier grass with increased feed value. 

## 4. Materials and Methods

### 4.1. Description of the Experimental Area and Planting Material

The study was conducted in Bishoftu, Ethiopia (008°4702000 N and 038°5901500 E) from 2018 to 2020. The altitude of Bishoftu is 1890 m.a.s.l., with an Alfisol soil type. A panel of 84 Napier grass genotypes sourced from the ILRI forage gene bank were used for the study (Table A1).

### 4.2. Field Trial Set Up

The trial was set up using a partially replicated (P-rep) design in four blocks. The field planting and growing conditions are as described in a previous report [42]. Land preparation, planting, weeding, harvesting and related management practices were uniformly applied across all plots. Inorganic fertilizer, urea and di ammonium phosphate (DAP) (50:50) were applied at a rate of 6.2 g/plant in each of the dry and wet seasons. The stress treatments, defined as moderate water stress (MWS) with 20% soil volumetric water content and severe water stress (SWS) with 10% soil volumetric water content, were imposed in the dry seasons (November to May), as indicated in our previous report [42] (Figure A3). As indicated in a report by [4], genotypes with drought stress tolerance characteristics also need to possess a high yield potential in non-stress environments to maximize annual yield; hence, genotypes were also evaluated during the rainy season (wet), which runs from June to September. The soil moisture content of the field plots was monitored using a Delta soil moisture probe (HD, UK) (Figure A5). In addition, a weather station (Speck ware technologies) was installed to monitor the daily climatic variables [4]. A composite soil sample from each replication was collected at the start and end of the experiment using an auger from a depth of up to 40 cm. The physical and chemical characteristics of the analyzed soil in the blocks are presented in Table 6.

### 4.3. Data Collection

After the genotypes were well established, the plants were harvested at about 5 cm above ground level after every eight weeks of regrowth, resulting in a total of 12 harvests, conducted between June 2018 and May 2020. In each harvest, morphological, physiological, agronomic and feed quality traits were collected. Three randomly selected plants were used for all trait measurements described below. 

Morphological traits: Plant height (PH—in cm), leaf length (LL—in cm), leaf width (LW—in mm), stem thickness (ST—in mm), tiller number (TN) and internode length (IL—in cm). PH was measured from the base to the tip of a randomly selected plant tiller, the same tiller was used to measure ST and IL, LL and LW were measured from the third leaf from the top of the plant. 

Agronomic traits: Total fresh weight (TFW), total dry weight (TDW) and leaf-stem ratio (LSR) were measured as described previously [13,43].

Physiological traits: Water use efficiency (WUE) was calculated by dividing the total dry weight per plant by the total volume of irrigated water applied per plant during the dry seasons [19]. Chlorophyll fluorescence (Fv/Fm) and performance index (PI) measurements were conducted on fully expanded leaves using a plant efficiency analyzer (Handy PEA; Hansatech Instrument Ltd., Lynn, UK) in the morning hours. The measured leaves were dark-adapted for 20 min before the measurement. The measurement consisted of a single strong 1 s light pulse (3000 L/mol photons m^−2^ s^−1^). 

Feed quality traits: In each harvest, the oven-dried whole plant, leaf and stem biomass fractions were ground separately for feed quality analysis. The feed quality traits for each component were measured as indicated by [13]; the traits measured were: neutral detergent fiber (NDF), acid detergent fiber (ADF), acid detergent lignin (ADL), organic matter (OM), crude protein (CP), in vitro organic matter digestibility (IVOMD) and metabolizable energy (Me). The feed quality traits are corrected and expressed on a percentage of dry matter (DM) basis, except for Me, which is expressed as MJ/Kg DM. The annual CP yield (CPY) was derived from the annual CP content and TDW yield by multiplying the annual CP by the annual TDW.

### 4.4. Data Analysis

An averaged data value per trait per genotype was generated in each of the three soil water regime conditions (MWS, SWS and Wet). There were six harvests for the MWS and SWS treatments, each with two replicates; however, in the wet season, as the soil moisture was similar across all blocks (Figure A3), the data from all four blocks were used for the six harvests. Hence, twelve harvests were used in this case. The averaged values for all traits were corrected for spatial variation using spatial analysis, assessed as described in a previous study [11,13].

The fitted data were used for the analysis described below. The values of the trait data were checked for normality using the bartlett test for heterogeneity. Statistical analysis was conducted using analysis of variance (ANOVA) in the GenStat software (version 19, VSN international Ltd., Hemel Hempstead, UK) [43] to determine the significance of the main effects and the interactions using the model:(1)$$Y_{\mathrm{ijk}}=\mu +G_{i}+T_{k}+H_{k}+\left(G_{i}{\;*\;T}_{\mathrm{ij}}\right)+\left(G_{i}{\;*\;H}_{\mathrm{ik}}\right)+\left(T_{j}{\;*\;H}_{\mathrm{jk}}\right)+\left(G_{i}{\;*\;T}_{j}{\;*\;H}_{k}\right)+\varepsilon_{\mathrm{ijk}}$$
where Y_ijk_ is the response during each season, μ = overall mean, G_i_ = effect of the ith genotype, T_j_ = effect of the jth treatment in the dry season (MWS/SWS), H_k_ = effect of the kth harvest period, G_i_ ∗ T_ij_ = the interaction of ith genotype and jth treatment, G_i_ ∗ H_ik_ = the interaction of ith genotype and kth harvest, T_j_ ∗ H_jk_ = the interaction of the jth treatment and kth harvest, G_i_ ∗ T_j_ ∗ H_k_ = three-factor interaction among genotypes, treatments and harvests and ε_ijk_ = the residual error. The least significant difference (LSD), for the comparison of the mean values of the traits, was employed to compare genotypes for traits with significant differences. The genetic parameters, genotypic coefficient of variation (GCV) and phenotypic coefficient of variation (PCV) were estimated using the formulae [44]: (2)$$\mathrm{GCV}=\frac{\sqrt{{\sigma g}^{2}}}{X}\times 100$$
(3)$$\mathrm{PCV}=\frac{\sqrt{{\sigma p}^{2}}}{X}\times 100$$
where GCV = genotypic coefficient of variation, PCV = phenotypic coefficient of variation, σ^2^g = genotypic variance, σ^2^p = phenotypic variance and X = grand mean. The broad-sense heritability (H^2^) for the traits was captured using the equation [25]:(4)$$H^{2}=\frac{{\sigma g}^{2}}{\left({\sigma g}^{2}+{\sigma e}^{2}\right)}$$
where σ^2^g and σ^2^e are the variance components for the genotype effect and the residual error, respectively.

To categorize genotypes in clusters in both the MWS and SWS of the dry season, a hierarchical co-cluster algorithm was employed using the R package ‘cluster’ version 2.1.2 [45]. The visualization of the cluster analysis was prepared using DeltaGen 3_1 [46]. 

Principle component analysis (PCA) based on the correlation matrix was performed using the packages ggplot2, factoextra and FactoMiner [47] to identify influential traits for selection. The Eigen values, latent vectors and PCA biplot were extracted from the PCA. PCA biplots were plotted separately for MWS and SWS using DeltaGen 3_1 [46] to show the relationships among the studied genotypes based on recorded traits. Principle component analysis (PCA) based on the correlation matrix was performed using the packages ggplot2, factoextra and FactoMiner [48] to identify influential traits for selection. The Eigen values, latent vectors and PCA biplot were extracted from the PCA. PCA biplots were plotted separately for MWS and SWS using DeltaGen 3_1 [46] to show the relationships among the studied genotypes based on recorded traits. 

For the graphical presentation of the mean values of traits, bar and line graphs were used. The relative magnitude of change in the trait values due to the two stress conditions, MWS and SWS, was presented as a radar plot. The graphical presentations of the bar graph, line graph and radar plots were prepared using Microsoft office software.

To select high-yielding genotypes growing under stressed conditions, a stress tolerance index (STI) was calculated using the following formula [47]:(5)$$\mathrm{STI}=\frac{\left(Y_{p}\;\times \;Y_{s}\right)}{\left({\mathrm{Xp}}^{2}\right)}$$
where Ys = total dry weight (TDW) of test genotypes growing under MWS or SWS conditions; Yp = total dry weight (TDW) and Xp = mean TDW of genotypes growing under the wet condition.

Principal component analysis (PCA) and hierarchical clustering were generated using Manhattan distance to classify the tolerance ranking via the average linkage method, using STI, WUE and TDW for both MWS and SWS conditions separately.

The main additive effects and multiplicative interaction (AMMI) analysis model was applied for MWS, SWS and wet conditions using the AMMI function in the R package Agricole [49] separately for 84 Napier grass genotypes (G) and 6 harvest time points (Environments = E) as additive effects and genotype environment interaction (GEI) as a multiplicative term. The AMMI analysis first fits additive effects for host genotypes and environments by the usual additive ANOVA procedure and then fits multiplicative effects for G × E (genotype × environment) by PCA.
(6)$${\;Y}_{\mathrm{ij}}=\mu +G_{i}+E_{j}+\sum_{k}^{n}\lambda k\alpha_{i}{k\gamma }_{j}k+e_{\mathrm{ij}}$$
where Y_ij_ is the TDW of the ith genotype in the jth environment, μ is the grand mean, G_i_ and E_j_ are the ith genotypic effect and jth environment effect, respectively, λk is the square root of the eigenvalue of the PCA axis k and α_i_k and γ_j_k are the principal component scores for the PCA axis k of the ith genotype and the jth environment.

The AMMI stability value (ASV) was computed as described in [50] using GenStat V19. Smaller ASV scores indicate a more stable genotype characteristic across environments.

The yield stability index was also calculated using the sum of the ranking based on yield and ranking based on the ASV value.
(7)YSI=RASV+RY
where RASV is the rank of the genotypes based on the ASV value; RY is the rank of the genotypes based on TDW across environments/harvests (RY). YSI incorporates both the mean yield and stability in a single criterion. Low values of both parameters show desirable genotypes with a high mean yield and stability.

To describe the magnitude of the relationships among agro-morphological and feed quality traits, Pearson’s correlation coefficients (*r*) were calculated separately for the dry and wet season conditions using GenStat (version 19, VSN international Ltd., Hemel Hempstead, UK) [43].

To evaluate the genotypes based on annual TDW and CP, cumulative TDW and CP yields across harvests were computed. The cumulative annual TDW was obtained by adding the mean TDW of the wet and dry season harvests for each genotype. The cumulative CP yield was obtained by adding the mean CP of wet and dry season harvests and multiplying by the respective cumulative TDW.

## 5. Conclusions

The field evaluation of 84 Napier grass genotypes sourced from the ILRI gene bank using moderate and severe soil moisture stress treatments elucidated significant variation in terms of agro-morphological, physiological and forage quality traits. The observed variations in phenotypic performance and the significant difference among genotypes in terms of measured traits indicated the possibility of developing varieties with improved water use efficiency and adaptability to drought stress environments. In general, the growth and development of most genotypes declined under SWS compared to MWS conditions. However, the genotypes that performed well in terms of WUE and TDW showed some level of consistency in both MWS and SWS conditions. For example, under MWS, genotypes 16819, 16803, 16839, BAGCE 30 and 16811 produced the highest forage yield; similarly, under SWS, the most productive genotypes were 16819, CNPGL-37-5, 16839, BAGCE 100 and BAGCE 30. In terms of forage quality traits such as crude protein content, high biomass-producing genotypes had low CP per kg of dry matter, but the total CP yield (CPY) of the high biomass-producing genotypes was also high, indicating the possibility of developing high-feed-quality Napier grass varieties under drought stress environments.

## Figures and Tables

**Figure 1 plants-11-02549-f001:**
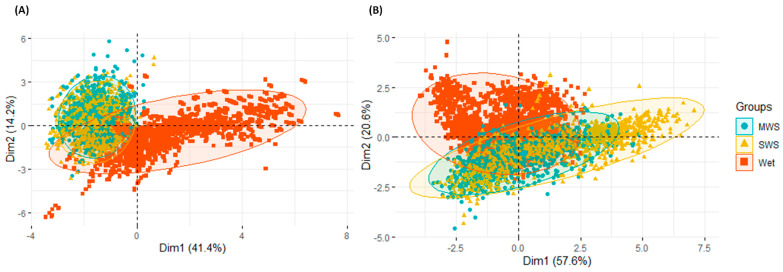
Principal component analysis of the genotypes grouped by: (**A**) agronomic traits and (**B**) feed quality traits, using pooled data from all harvests. Genotypes are grouped according to if they were grown under moderate water stress (MWS) (blue color), severe water stress (SWS) (yellow color) or rainy season (wet) (red color) conditions.

**Figure 2 plants-11-02549-f002:**
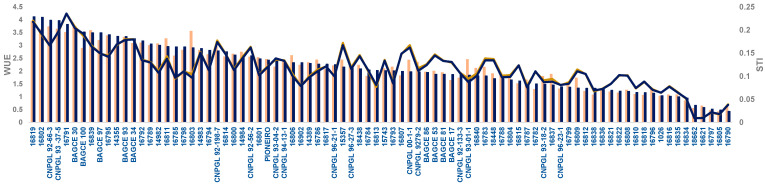
Water use efficiency (WUE-Bar graph) and stress tolerance index (STI-Line graph) profile of Napier grass genotypes grown under moderate water stress (light brown) or severe water stress (blue black) conditions. The order of genotypes from left to right is based on WUE rank (1 to 84) under severe water stress.

**Figure 3 plants-11-02549-f003:**
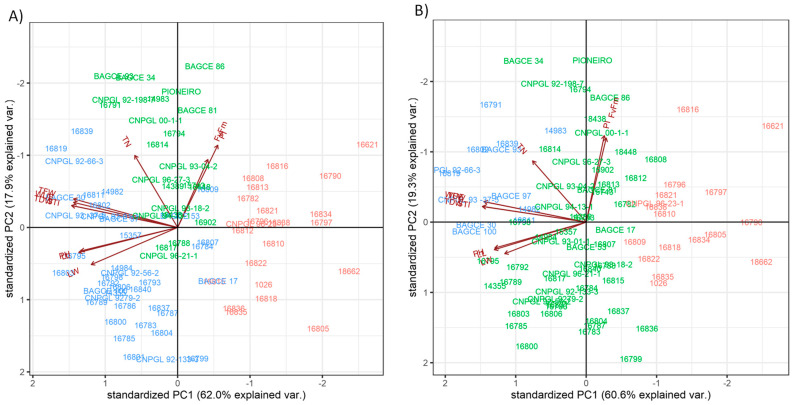
PCA biplots of morphological, physiological and agronomic traits of Napier grass genotypes growing under: (**A**) moderate water stress and (**B**) severe water stress. The color of genotypes indicates different cluster groups from the pattern analysis. The angles between the vectors indicate the degree of correlation between traits. Plant height (PH), leaf length (LL), leaf width (LW), chlorophyll fluorescence (Fv/Fm), performance index (PI), tiller number (TN), total fresh weight (TFW), total dry weight (TDW), water use efficiency (WUE) and stress tolerance index (STI).

**Figure 4 plants-11-02549-f004:**
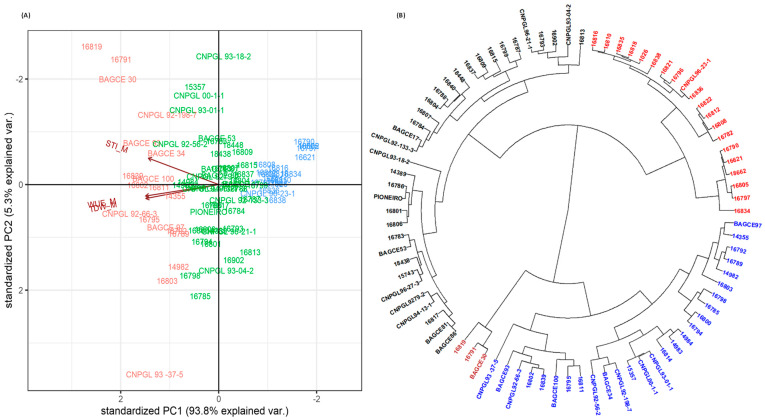
Grouping by drought stress response of Napier grass genotypes, based on STI, WUE and TDW, under moderate water stress using: (**A**) PCA and (**B**) cluster analysis. Both analyses discriminate highly tolerant, moderately tolerant and susceptible genotypes. Water use efficiency (WUE), total dry weight (TDW) and stress tolerance index (STI).

**Figure 5 plants-11-02549-f005:**
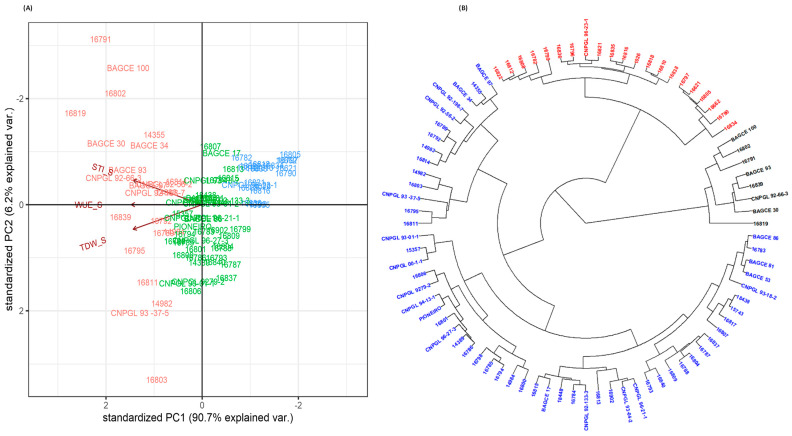
Grouping by drought stress response of Napier grass genotypes, based on STI, WUE and TDW, under severe water stress using: (**A**) PCA and (**B**) cluster analysis. Both analyses discriminate highly tolerant, moderately tolerant and susceptible genotypes. Water use efficiency (WUE), total dry weight (TDW) and stress tolerance index (STI).

**Figure 6 plants-11-02549-f006:**
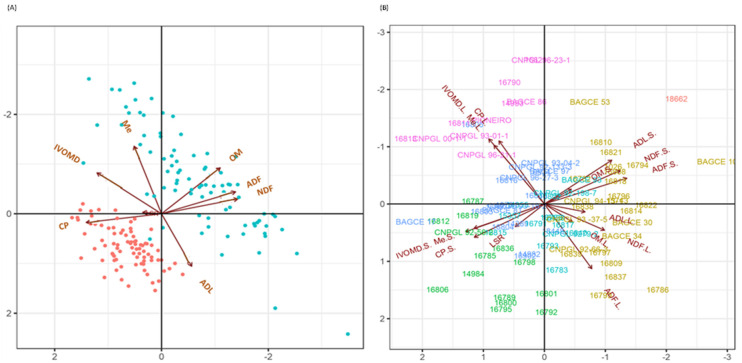
PCA biplots of seven feed quality traits and leaf-to-stem ratio for leaf and stem samples of Napier grass genotypes under the wet season: (**A**) Distribution of leaf (red dots) and stem (blue dots) tissue samples for feed quality traits; (**B**) Distribution of genotypes for feed quality traits in leaf and stem samples. The color of genotypes indicates different cluster groups. The angles between the vectors indicate positive and negative correlations among traits. Neutral detergent fiber (NDF); acid detergent fiber (ADF); acid detergent lignin (ADL); organic matter (OM); crude protein (CP); in vitro organic matter digestibility (IVOMD); metabolizable energy (ME). The suffixes L and S on the feed quality traits indicate leaf samples and stem samples, respectively.

**Table 1 plants-11-02549-t001:** Combined ANOVA for agro-morphological traits of 84 Napier grass genotypes tested under rainy (wet) and dry (dry) season conditions.

Sources of Variation/Traits	Season	Genotype (G)	Treatment (T)	Harvest (H)	G X T	G X H	T X H	G X T X H	CV %
PH	Wet	<0.001	NS	<0.001	NS	<0.001	NS	NS	2.9
	Dry	<0.001	0.05	<0.001	0.02	<0.001	<0.001	<0.001	11.2
LW	Wet	<0.001	NS	<0.001	NS	<0.001	NS	NS	1.6
	Dry	<0.001	0.04	<0.001	NS	<0.001	<0.001	NS	9.4
LL	Wet	<0.001	NS	<0.001	NS	<0.001	<0.001	NS	2
	Dry	<0.001	0.03	<0.001	<0.001	<0.001	<0.001	<0.001	6.1
IL	Wet	<0.001	NS	<0.001	NS	<0.001	<0.001	NS	49.8
	Dry	-	-	-	-	-	-	-	
ST	Wet	<0.001	NS	<0.001	NS	<0.001	NS	NS	2.2
	Dry	-	-	-	-	-	-	-	
TN	Wet	<0.001	NS	<0.001	NS	<0.001	<0.001	NS	12.3
	Dry	<0.001	<0.001	<0.001	<0.001	<0.001	<0.001	<0.001	5.2
Fv/Fm	Wet	<0.001	NS	<0.001	NS	<0.001	<0.001	NS	5.6
	Dry	<0.001	0.04	<0.001	NS	<0.001	<0.001	NS	2.6
PI	Wet	<0.001	NS	<0.001	NS	<0.001	<0.001	NS	9.9
	Dry	<0.001	0.07	<0.001	<0.001	<0.001	<0.001	<0.001	20.7
TFW	Wet	<0.001	NS	<0.001	NS	<0.001	<0.001	NS	6.2
	Dry	<0.001	<0.001	<0.001	<0.001	<0.001	<0.001	<0.001	13.3
TDW	Wet	<0.001	NS	<0.001	NS	<0.001	<0.001	NS	9.4
	Dry	<0.001	0.05	<0.001	<0.001	<0.001	<0.001	<0.001	17.8
LSR	Wet	<0.001	NS	<0.001	NS	<0.001	<0.001	NS	54.6
	Dry	-	-	-	-	-	-	-	
WUE	Wet	-	-	-	-	-	-	-	
	Dry	<0.001	0.05	<0.001	<0.001	<0.001	NS	<0.001	13.4

Plant height (PH), Leaf width (LW), Leaf length (LL), Internode length (IL), Stem thickness (ST), Tiller number (TN), Chlorophyll fluoresce (Fv/Fm), Performance index (PI), Total fresh weight (TFW), Total dry weight (TDW), Leaf-to-stem ratio (LSR), Water use efficiency (WUE), Coefficient of variation (CV). Values indicate the significance probability level; not significant (NS). The dash sign (-) indicate data not recorded for the trait in the specified season.

**Table 2 plants-11-02549-t002:** Variations and heritability in agro-morphological traits of Napier grass genotypes grown under wet and dry season conditions—either exposed to moderate water stress (MWS) or severe water stress (SWS)—for two years.

Traits	Growing Conditions	Mean	Range	PCV%	GCV%	H^2^ %
PH	Wet	55.49	5.01–115.4	51.11	23.13	45.25
	MWS	12.88	0.50–28.05	34.46	22.11	64.16
	SWS	12.5	1.77–26.1	34.01	20.37	59.9
LL	Wet	79.61	14.19–143.1	27.16	18.26	67.21
	MWS	42.01	0.1–72.71	32.78	28.01	85.43
	SWS	42.79	8.45–69.02	29.05	23.75	81.76
LW	Wet	26.2	9.85–39.44	22.29	19.38	86.92
	MWS	18.13	2.5–30.36	22.96	15.06	65.62
	SWS	19.01	5.07–30.67	21.73	14.11	64.95
Fv/Fm	Wet	0.74	0.63–0.81	4.93	2.46	50.02
	MWS	0.74	0.55–0.85	4.82	2.06	42.62
	SWS	0.73	0.54–0.82	4.9	2.11	43.15
PI	Wet	4.37	0.70–12.69	44.59	19.11	42.85
	MWS	3.66	0.44–11.43	40.13	19.88	49.54
	SWS	3.45	0.01–21.5	41.56	18.95	45.6
TN	Wet	62.97	2.07–262.5	64.24	51.01	79.4
	MWS	134.3	4.08–494.8	59.52	39.27	65.97
	SWS	131.3	6.00–439.5	62.07	41.94	67.57
TFW	Wet	43.67	0.13–184.4	89.21	55.3	61.99
	MWS	5.31	0.01–20.08	68.85	40.59	58.95
	SWS	5.13	0.01–20.08	71.32	42.02	58.92
TDW	Wet	9.83	0.10–34.17	81.66	58.58	71.73
	MWS	1.45	0.001–6.22	68.3	42.89	62.79
	SWS	1.34	0.001–6.22	70.32	40.44	57.51
WUE	MWS	2.17	0.01–10.16	79.84	38.45	48.15
	SWS	2.16	0.01–10.16	81.5	41.22	50.58
LSR	Wet	5.15	0.90–55.95	115.36	68.93	59.76
ST	Wet	14.21	3.03–176.2	62.94	37.62	59.77
IL	Wet	24.3	10.08–53.03	37.79	17.09	45.24

Plant height (PH), Leaf width (LW), Leaf length (LL), Internode length (IL), Stem thickness (ST), Tiller number (TN), Chlorophyll fluoresce (Fv/Fm), Performance index (PI), Total fresh weight (TFW), Total dry weight (TDW), Leaf-to-stem ratio (LSR), Water use efficiency (WUE), Genotypic coefficient of variation (GCV), Phenotypic coefficient of variation (PCV), Heritability in a broad sense (H^2^).

**Table 3 plants-11-02549-t003:** Combined ANOVA for seven feed quality traits of Napier grass genotypes grown under rainy season and dry season conditions across six harvests.

Sources of Variation/Traits	Season	Genotype (G)	Treatment (T)	Harvest (H)	G X T	G X H	T X H	G X T X H	CV %
NDF	Wet	<0.001	NS	<0.001	<0.001	<0.001	<0.001	<0.001	1.3
	Dry	<0.001	<0.001	<0.001	<0.001	<0.001	<0.001	<0.001	0.5
ADF	Wet	<0.001	NS	<0.001	<0.001	<0.001	<0.001	<0.001	1.5
	Dry	<0.001	0.04	<0.001	<0.001	<0.001	<0.001	<0.001	3.3
ADL	Wet	<0.001	NS	<0.001	<0.001	<0.001	<0.001	<0.001	2.6
	Dry	<0.001	0.002	<0.001	<0.001	<0.001	<0.001	<0.001	1.9
OM	Wet	<0.001	NS	<0.001	<0.001	<0.001	<0.001	<0.001	0.2
	Dry	<0.001	NS	<0.001	<0.001	<0.001	<0.001	<0.001	1.1
CP	Wet	<0.001	NS	<0.001	<0.001	<0.001	<0.001	<0.001	3.5
	Dry	<0.001	<0.001	<0.001	<0.001	<0.001	<0.001	<0.001	6.8
IVOMD	Wet	<0.001	NS	<0.001	<0.001	<0.001	<0.001	<0.001	0.9
	Dry	<0.001	<0.001	<0.001	<0.001	<0.001	<0.001	<0.001	1
Me	Wet	<0.001	NS	<0.001	<0.001	<0.001	<0.001	<0.001	0.9
	Dry	<0.001	0.04	<0.001	<0.001	<0.001	<0.001	<0.001	1.8

Neutral detergent fiber (NDF); acid detergent fiber (ADF); acid detergent lignin (ADL); organic matter (OM); crude protein (CP); in vitro organic matter digestibility (IVOMD); metabolizable energy (ME); coefficient of variation (CV). Values indicate the significance probability level; not significant (NS).

**Table 4 plants-11-02549-t004:** Variations and heritability of the feed quality traits of 84 Napier grass genotypes grown under rainy season (wet) and dry season conditions—either exposed to moderate water stress (MWS) or severe water stress (SWS)—for two years.

Traits	Growing Condition	Mean	Range	PCV%	GCV%	H^2^ %
NDF	Wet	67.58	58.1–78.59	4.89	1.37	28.06
	MWS	63.75	57.29–69.59	3.18	1.9	59.68
	SWS	62.86	56.27–69.59	3.65	1.51	41.31
ADF	Wet	41.25	33.72–48.1	7.37	2.17	29.43
	MWS	37.5	29.65–45.78	8.16	2.31	28.32
	SWS	34.62	25.15–45.00	12.51	2.35	18.81
ADL	Wet	3.88	1.93–3.79	28.22	3.89	13.8
	MWS	2.81	2.05–3.91	11.87	4.04	34
	SWS	2.69	1.93–3.79	10.4	4.76	45.72
OM	Wet	82.68	72.69–95.72	1.59	0.72	45.62
	MWS	81.7	72.11–87.72	2.76	1.33	48.31
	SWS	82.13	72.69–95.72	3.28	1.38	41.91
CP	Wet	12.07	5.05–24.3	19.74	4.69	23.76
	MWS	10.94	4.53–20.83	27.99	12.26	43.82
	SWS	13.86	5.05–24.3	29.38	9.76	33.24
IVOMD	Wet	55.15	50.38–69.27	4.53	1	22.12
	MWS	56.24	50.45–65.06	5.19	1.49	28.68
	SWS	58.7	50.38–69.27	7.83	0.23	2.95
Me	Wet	7.65	6.74–9.58	4.31	0.66	15.34
	MWS	7.89	6.68–9.15	6.23	0.64	10.28
	SWS	8.15	6.74–9.58	8.56	0.39	4.54

Neutral detergent fiber (NDF); acid detergent fiber (ADF); acid detergent lignin (ADL); organic matter (OM); crude protein (CP); in vitro organic matter digestibility (IVOMD); metabolizable energy (ME); genotypic coefficient of variation (GCV); phenotypic coefficient of variation (PCV); heritability in a broad sense (H^2^).

**Table 5 plants-11-02549-t005:** Annual total dry weight and crude protein yield of the 84 Napier grass genotypes.

Genotype	Moderate Water Stress		Severe Water Stress		Genotype	Moderate Water Stress		Severe Water Stress	
	Annual TDW t/ha	Annual CPY t/ha	Annual TDW t/ha	Annual CPY t/ha		Annual TDW t/ha	Annual CPY t/ha	Annual TDW t/ha	Annual CPY t/ha
1026	18.9p	1281.06l	17.99b	1448.47hi	16816	22.8lmn	1570.71hi	22.05yza	1863.46abcdef
14355 **	48.48ij	3240.65ghij	48.21fg	3648.34efg	16817	35.68tuvw	2300.43uvw	34.07opq	2456.48rst
14389	29.28cdefg	2154.83wxy	27.75uv	2172.14uvwxy	16818	25.47ijk	1640.72ghi	25.13wx	1931zabcd
14982	31.91yzab	2375.85stu	30.83rst	2343.17stu	16819 **	63.57b	3893.9c	62.43b	4430.8b
14983	43.95k	3423.47ef	43.31h	3862.63de	16821	24.41klm	1613.06ghi	23.56xy	1719.4efg
14984	41.02lmno	2663.53p	39.33jk	2877.39lm	16822	29.58bcdefg	1901.39bcd	29.1tuv	1907.54abcde
15357 **	49.97hi	3342.45efgh	47.66fg	3572.8fg	16834	15.26q	1111.38m	14.43c	1177.9j
16621	2.9t	164.84p	2.66f	201.87lm	16835	19.25p	1377.79kl	18.14b	1413.72i
16782	31.53yzabc	2139.68wxyz	31.71pqrs	2080.56vwxyza	16836	21.56no	1525.72ij	20.33a	1778.28cdef
16783	40.43mno	2511.52qrs	38.16kl	2647.38nopqr	16837	27.5ghi	1835.66cde	24.92x	1908.07abcde
16784	32.75xyza	2369.07tu	32.33pqr	2691.07nopq	16838	19.35op	1372.03kl	19.98ab	1320.31ij
16785	27.76efghi	1515.86ijk	27.47vw	1808.73bcdef	16839**	48.09ij	3199.29hijk	46.91fg	3567.88fgh
16786	33.3wxyz	2025.89yzab	31.6rs	2260.51tuvw	16840	30.84zabcd	2036.14yzab	28.66tuv	2081.19vwxyza
16787	23.34klmn	1522.19ij	21.21za	1668.73fgh	16902	23.33klmn	1578.78hi	21.98yza	1739.71defg
16788	29.66bcdef	1978.48zab	27.61uv	2080.21vwxyza	18438 *	42.39klmn	2958.7no	40.76ij	3718.42ef
16789	38.09pqrs	2313.41uv	36.9lmn	2619.82opqr	18448 *	40.05nop	2966.61no	38.12kl	3115.94jk
16790	11.64r	813.97n	10.45d	790.64k	18662	2.65t	151.77p	2.47f	139.56m
16791 **	68.05a	4429.1a	67.35a	5213.84a	15743	39.93nop	2530.31pqrs	37.49klmn	2501.03qrs
16792	39.54op	2200.51vwx	38.41kl	2494.38rs	BAGCE 100 **	54.21ef	3101.07jklmn	55.11d	3684.03ef
16793	25.56ijk	1585.26ghi	23.86xy	1669.11fgh	BAGCE 17 *	37.38qrst	3016.64lmn	37.57klm	3215.17ijk
16794	33.46wxy	2333.49tuv	31.63qrs	2705.3mnopq	BAGCE 30 **	59.94c	4130.07b	58.37c	4466.59b
16795	42klmn	2632.63pq	40.48ij	2828.19lmn	BAGCE 34 **	52.51fg	3145.92ijklm	51.03e	3750.61def
16796	21.16nop	1393.36jkl	19.71ab	1534.64ghi	BAGCE 53 *	42.74klm	2976.08no	41.28hij	3348.2hij
16797	6.28s	464.86o	6.11e	421.85l	BAGCE 81 *	39.02opqr	2943.12no	37.75kl	3163.11jk
16798	31.89yzab	1966.69abc	31.62qrs	2247.3tuvw	BAGCE 86	36.58rstu	2600.97pqr	35.22mno	2518.56pqrs
16799	25.1jkl	1693.38fgh	23.87xy	1737.6defg	BAGCE 93 **	52.27fg	3818.98cd	50.94e	4469.65b
16800	33.72vwxy	2024.91yzab	31.95pqrs	2180.03uvwxy	BAGCE 97 *	43.35k	3256.66ghij	42.43hi	3631.1efg
16801	30.26bcd	1829.93cdef	28.89tuv	1987.33xyzabc	CNPGL 00-1-1 **	48.86ij	3697.89d	45.97g	3520.77fgh
16802 **	55.11de	3342.67efgh	55.47d	3966.85cd	CNPGL 92-133-3	30.88zabcd	2165.94wxy	30.88rst	2711.36mnopq
16803	30.69abcd	2028.23yzab	27.25vw	1918.03zabcde	CNPGL 92-198-7 **	51.64gh	3381.83efg	49.35ef	4169.44c
16804	30.09bcdef	2202.14vwx	27.73uv	2219.09uvwx	CNPGL 92-56-2 **	47.63j	3156.48ijkl	46.1g	3541.91fgh
16805	5.1s	400.24o	4.79e	297.49lm	CNPGL 92-66-3 **	48.33ij	3053.45klmn	47.38fg	3422.05ghi
16806	30.59bcd	2017.28yzab	28.36uv	1957.29yzabcd	CNPGL 9279-2	34.22vwx	2208.15vwx	31.57rs	2260.57tuv
16807	43kl	3297.35fghi	42.26hi	3201.01ijk	CNPGL 93 -37-5 **	56.71d	3479.82e	55.75d	3967.35cd
16808	29.74bcdef	2008.26yzab	28.74tuv	2152.11uvwxyz	CNPGL 93-01-1	27.66fghi	2138.5xyz	24.67x	2030.17wxyzab
16809	33.84vwxy	2370.2stu	30.95rst	2113.2uvwxyz	CNPGL 93-04-2 *	39.51opq	2952.96no	39.58jk	3160.9jk
16810	21.61no	1582.54ghi	21.09za	1494.81hi	CNPGL 93-18-2	26.6hij	2022.46yzab	24.62x	1831.22bcdef
16811 *	41.69klmn	3331.09efgh	39.15jk	3270.85ij	CNPGL 94-13-1	38.99opqr	2993.54mn	38.05kl	2735.53mnop
16812	30.21bcde	2467.96rst	29.99stu	2514.4pqrs	CNPGL 96-21-1	28.93defgh	2094.59xyza	28.57tuv	2251.7tuvw
16813	22.46mn	1741.49defg	22.98xyz	1865.04abcdef	CNPGL 96-23-1	23.5klmn	1698.31efgh	23.03xyz	1780.96cdef
16814	42.99kl	2600.77pqr	42.19hi	2983.3kl	CNPGL 96-27-3	34.53uvwx	2374.71stu	33.08opqr	2345.12stu
16815	36.1stuv	2825.58o	35.05no	3024.79kl	Pioneiro	35.18tuvw	2558.97pqr	34.15op	2797.11lmno

Annual total dry weight (TDW) is the mean of the two-year total dry weight of Napier genotypes grown under moderate water stress (MWS) or severe water stress conditions (SWS) in both the dry and wet seasons; annual crude protein yield (CPY) is the product of the annual TDW and the annual crude protein (CP) of Napier grass genotypes grown under moderate water stress (MWS) or severe water stress (SWS) conditions in both the dry and wet seasons; means with different letters in the same column are significantly different at *p* < 0.05. * Indicates top CPY-producing genotypes under MWS and * indicates top CPY-producing genotypes under SWS.

**Table 6 plants-11-02549-t006:** Soil chemical properties at a depth of 0–40 cm in the experimental blocks at the start and end of the study.

Soil Chemical Properties		Block			
	Year	1	2	3	4
Phosphorus (ppm)	2018	14.06	12	11.27	10.69
	2020	20.93	24.92	14.32	23.44
Potassium (%)	2018	335.27	354.03	339.2	279.78
	2020	294	320.25	455.52	367.47
Organic carbon (C)	2018	1.11	1.07	1	0.99
	2020	1.16	1.14	1.06	1.01
Total nitrogen (N)	2018	0.08	0.09	0.09	0.08
	2020	0.1	0.1	0.09	0.09
C:N	2018	13.94	11.85	11.6	13.18
	2020	11.49	11.75	11.67	11.03
Cation exchange capacity	2018	29.22	28.8	23.75	26.92
	2020	28.09	27.27	25.59	21.5
PH	2018	7.26	7.22	7.26	7.12
	2020	8.82	8.56	8.7	8.41

Organic-carbon-to-nitrogen ratio (C:N), parts per million (ppm).

## Data Availability

Not applicable.

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
