# Peer review of "Productivity and Feed Quality Performance of Napier Grass (Cenchrus purpureus) Genotypes Growing under Different Soil Moisture Levels"

_plants, 2022, doi:10.3390/plants11192549_

Round 1

Reviewer 1 Report

The authors assessed the drought resistance of 84 genotypes of Napier grass under field water stress conditions of moderate and severe water stress in the dry season and rainfed in the wet season. The results showed that growth and productivity of all genotypes decreased under severe drought conditions compared to moderate drought conditions. The phenotype level is completely covered, and it is expected that in the future, it may be combined with the transcription level to discover more stress resistance genes. The following recommendations are made for this study.

1. The content of the Result is too much, and the appropriate combination.

2. There are many Figures, and some of them can be considered as Supplementary Figure.

Author Response

Response letter attached

Reviewer 2 Report

I suggest improving the sub-component 2.9. Biomass Yield Stability across Harvests. 

2. In the Feed Quality methodology give a details description of Feed Trait analysis.  Explain why leaf contents differ from stem feed quality. This is a very interesting postulate. Did you consider which ontogenetic stage is most visible: active green biomass accumulation; seed maturation etc. 

3. If authors have any data on testing biomass, collected under different stress conditions to feed animals. Animal perception. If not described in the Discussion chapter. 

Author Response

Response letter attached
